# Effective Low-Energy Hamiltonians and Unconventional Landau-Level Spectrum of Monolayer C_3_N

**DOI:** 10.3390/nano12244375

**Published:** 2022-12-08

**Authors:** Mohsen Shahbazi, Jamal Davoodi, Arash Boochani, Hadi Khanjani, Andor Kormányos

**Affiliations:** 1Department of Physics, Faculty of Science, University of Zanjan, Zanjan P.O. Box 45195-313, Iran; jdavoodi@znu.ac.ir; 2Department of Physics, Kermanshah Branch, Islamic Azad University, Kermanshah P.O. Box 671791-7855, Iran; arash_bch@yahoo.com; 3Quantum Technological Research Center (QTRC), Science and Research Branch, Islamic Azad University, Tehran P.O.Box 14515-755, Iran; 4Department of Physics, University of Tehran, Tehran P.O. Box 14395-547, Iran; hadikhanjani@gmail.com; 5Department of Physics of Complex Systems, Eötvös Loránd University, 1117 Budapest, Hungary

**Keywords:** 2D materials, electronic properties, Landau levels

## Abstract

We derive low-energy effective 
k·p
 Hamiltonians for monolayer 
$C_{3}$
N at the 
Γ
 and *M* points of the Brillouin zone, where the band edge in the conduction and valence band can be found. Our analysis of the electronic band symmetries helps to better understand several results of recent ab initio calculations for the optical properties of this material. We also calculate the Landau-level spectrum. We find that the Landau-level spectrum in the degenerate conduction bands at the 
Γ
 point acquires properties that are reminiscent of the corresponding results in bilayer graphene, but there are important differences as well. Moreover, because of the heavy effective mass, *n*-doped samples may host interesting electron–electron interaction effects.

## 1. Introduction

Graphene [1] has received a great deal of attention due to its unique mechanical, electronic, thermal and optoelectronic properties [2,3,4]. However, having a zero band gap limited the applications of graphene in electronic nano-devices and motivated the search for atomically thin two-dimensional (2D) materials, which have a finite band gap. This led to the discovery of, for example, monolayer transition metal dichalcogenides [5,6,7], silicene [8,9], phosphorene [10,11], and germanene [12]. In recent years, compounds of carbon nitrides 
$C_{x}$

$N_{y}$
 have also become attractive 2D materials [13,14,15]. For example, graphitic carbon-nitride (g-
$C_{3}$

$N_{4}$
), which is a direct band gap semiconductor, has potential applications in photocatalysis and in solar energy conversion due to its strong optical absorption at visible frequencies [16,17]. Another carbon-nitride compound, two-dimensional crystalline 
$C_{3}$
N, has also been recently synthesized [18,19]. 
$C_{3}$
N is an indirect band gap semiconductor with an energy gap of 
0.39
 eV [19]. Moreover, it has shown favorable properties, such as high mechanical stiffness [20] and interesting excitonic effects [21,22]. In addition, its thermal conductivity properties have been investigated [20,23,24], and it has been predicted that the electronic, optical and thermal properties of monolayer 
$C_{3}$
N can be tuned by strain engineering [25,26].

In this work, we employ the 
k·p
 [27,28] approach in order to study the electronic properties of monolayer 
$C_{3}$
N. We obtain the materials specific parameters appearing in the 
k·p
 model from fitting it to density functional theory (DFT) band structure calculations. A similar methodology has been successfully used, e.g., for monolayers of transition metal dichalcogenides [29,30]. In particular, since the conduction band (CB) minimum and valence band (VB) maximum are located at the 
Γ
 and *M* points of the Brillouin zone, respectively, we obtain 
k·p
 Hamiltonians valid in the vicinity of these points. The insight given by the 
k·p
 model allows us to comment on certain optical properties as well. Moreover, we will also study the Landau-level spectrum of 
$C_{3}$
N, which, to our knowledge, has not been considered before.

This paper is organized as follows. In Section 2, we start with a short recap of the band structure obtained with the help of the density functional theory calculations. In Section 3, effective 
k·p
 Hamiltonians at 
Γ
 and *M* points are obtained, using symmetry groups and perturbation theory. Certain optical properties of this material are discussed in Section 4. In Section 5, the spectra of Landau levels for this material are calculated at the 
Γ
 and *M* points. Finally, our main results are summarized in Section 6.

## 2. Band Structure Calculations

The band structure of monolayer 
$C_{3}$
N has been calculated before at the DFT level of theory [25,31,32] and also using the GW approach [21,22,33]. The main effect of the GW approach is to enhance the band gap, and this does not affect our main conclusions below. To be self-contained, we repeat the band structure calculations at the DFT level. The schematics of the crystal lattice of single-layer 
$C_{3}$
N is shown in Figure 1. The lattice of 
$C_{3}$
N possesses P6/mmm space group with a planar hexagonal lattice, and the unit cell contains six carbon and two nitrogen atoms. We used the Wien2K package [34] to perform first-principles calculations based on density functional theory (DFT). For the exchange–correlation potential, we used the generalized gradient approximation [35]. The optimized input parameters, such as RKmax, lmax, and k-point, were selected to be 
8.5
, 10, and 
14×14×3
, respectively. The convergence accuracy of self-consistent calculations for the electron charge up to 
0.0001
 was chosen, and the forces acting on the atoms were optimized to 
0.1
 dyn/a.u. The optimized lattice constant is 
$a_{0}=4.86$
, in good agreement with previous studies [31,36].

The calculated band structure is shown in Figure 2. The conduction band minimum is located at the 
Γ
 point, while the valence band maximum can be found at the *M* of the BZ. Thus, at the DFT level, 
$C_{3}$
N is an indirect band gap semiconductor with a band gap of 
$E_{bg}=0.48$
 eV which is in good agreement with previous works [19,37]. We checked that the magnitude of the spin-orbit coupling is small at the band-edge points of interest and, therefore, in the following, we will neglect it. The main effect of spin-orbit coupling is to lift degeneracies at certain high-symmetry points and lines, e.g., the four-fold degeneracy of the conduction band at the 
Γ
 point would be split into two, two-fold degenerate bands.

## 3. Effective 
k·p
 Hamiltonians

We now introduce the 
k·p
 for the 
Γ
 point, where the band edge of the CB is located, and for the *M* point, where the band edge of the VB can be found.

### 3.1. 
Γ
 Point

The pertinent point group at the 
Γ
 point of the BZ is 
$D_{6h}$
. We obtained the corresponding irreducible representations of the nine bands around the Fermi level at the 
Γ
 point with the help of the Wien2k package. Using this information, one can then set up a nine-band 
k·p
 model along the lines of Ref. [30]; see Appendix A for details. Here, we only mention that there is no 
k·p
 matrix element between the VB and the degenerate CB, CB+1, which means that direct optical transitions are not allowed between these two bands. Since it is usually difficult to work with a nine-band Hamiltonian, we derive an effective low-energy Hamiltonian which describes the two (degenerate) conduction bands and the valence band. Using the Löwdin partitioning technique [38,39], we find that

(1)
$$\begin{array}{c} H_{\mathrm{eff}}^{\Gamma }=H_{0}^{\Gamma }+H_{k\cdot p}^{\Gamma }, \end{array}$$


(2)
$$\begin{array}{c} H_{0}^{\Gamma }=\left(\begin{array}{ccc} \varepsilon_{vb} & 0 & 0\\ 0 & \varepsilon_{cb} & 0\\ 0 & 0 & \varepsilon_{cb+1} \end{array}\right) \end{array}$$


(3)
$$\begin{array}{c} H_{k\cdot p}^{\Gamma }=\left(\begin{array}{ccc} \alpha_{1}q^{2} & 0 & 0\\ 0 & \left(\alpha_{2}+\alpha_{3}\right)q^{2} & -\alpha_{3}{\left(q_{+}\right)}^{2}\\ 0 & -\alpha_{3}{\left(q_{-}\right)}^{2} & \left(\alpha_{2}+\alpha_{3}\right)q^{2} \end{array}\right). \end{array}$$


Here, 
$\varepsilon_{cb}=\varepsilon_{cb+1}=0.386$
 eV and 
$\varepsilon_{vb}=-1.50$
 eV are band edge energies of the degenerate CB minimum and VB maximum. The wavenumbers 
$q_{x}$
, 
$q_{y}$
 are measured from the 
Γ
 point, 
$q_{\pm }=q_{x}\pm iq_{y}$
 and 
$q^{2}=q_{x}^{2}+q_{y}^{2}$
, and in 
$\alpha_{2}$
 we took into account the free electron term [29].

Note that there are no linear-in-
q
 matrix elements between the VB and the degenerate CB, CB+1 bands. In the higher order of 
q
, these bands do couple, but this is neglected in the minimal model given in Equation (Equation 3). The minimal model given in Equations (Equation 1)–(Equation 3) already captures an important property of the degenerate CB and CB+1 bands from the DFT calculations, which is that their effective masses are different. One finds from Equation (Equation 3) that the effective masses are 
$1/m_{cb}^{\Gamma }=\frac{2}{\hbar^{2}}\alpha_{2}$
 and 
$1/m_{cb+1}^{\Gamma }=\frac{2}{\hbar^{2}}\left(\alpha_{2}+2\alpha_{3}\right)$
. The material parameters 
$\alpha_{i}$
 can be obtained, e.g., by fitting these effective masses to the DFT band structure calculations. We find that 
$\alpha_{1}=51.18$
 eV
Å2
, 
$\alpha_{2}=17.68$
 eV
${Å}^{2}$
 and 
$\alpha_{3}=13.89$
 eV
${Å}^{2}$
. The corresponding effective masses at the 
Γ
 point are shown in Table 1.

### 3.2. **M** Point

Next we consider the *M* point, where the location of the VB maximum is. The relevant point group is 
$D_{2h}$
. Since this point group has only one-dimensional irreducible representation, one expects that there are no degenerate bands near the *M* point. This is in agreement with our DFT calculations; see Figure 2. Because of the dense spectrum in the conduction band, we start with a 13 bands 
k·p
 Hamiltonian (see Appendix B for details), and by projecting out the higher energy bands, we obtain an effective two-band model for the VB and the CB.

This effective model reads

(4)
$$H_{\mathrm{eff}}^{M}=H_{0}^{M}+H_{k\cdot p}^{M},$$


(5)
$$H_{0}^{M}=\left(\begin{array}{cc} \varepsilon_{vb} & 0\\ 0 & \varepsilon_{cb} \end{array}\right),$$


(6)
$$H_{k\cdot p}^{M}=\left(\begin{array}{cc} \frac{\hbar^{2}}{2m_{e}}q_{x}^{2}+\beta_{1}q_{y}^{2} & \gamma_{21}q_{x}\\ \gamma_{21}^{*}q_{x} & \beta_{2}q_{x}^{2}+\beta_{3}q_{y}^{2} \end{array}\right).$$


Here, 
$\varepsilon_{vb}=0.021$
 eV and 
$\varepsilon_{cb}=1.65$
 eV refer to band-edge energies of the VB and the CB, respectively, and the 
$q_{x}$
 direction is along the 
Γ−M
 line. We note that 
$H_{k\cdot p}^{M}$
 includes a free electron term as well [29]. It is interesting to note that 
$H_{k\cdot p}^{M}$
 has the same general form as the 
k·p
 model for the 
Γ
 point of phosphorene [40,41,42]. An important difference between the two cases, apart from the fact that the multiplicity of the 
Γ
 and *M* points is different, is that in the case of 
$C_{3}$
N, there is a saddle point in the dispersion at the *M* points, whereas in the case of phosphorene, the dispersion has a positive slope in every direction at the 
Γ
 point. The material parameters appearing in Equation (Equation 6) can be obtained by fitting the dispersion to the DFT band structure calculations. The range of fitting is 
0.5%
 of both the 
M−K
 and 
M−Γ
 directions. We find 
$\beta_{1}=-111.7$
 eV
Å2
, 
$\beta_{2}=-0.71$
 eV
Å2
, 
$\beta_{3}=125.3$
 eV
${Å}^{2}$
, and 
$\gamma_{21}=5.61$
 eVÅ. The corresponding effective masses are given in Table 1. One can indeed see that the effective masses have a different sign along the 
M−K
 and 
M−Γ
 directions.

## 4. Comments on the Optical Properties

Recently, Refs. [21,22] studied the optical properties of 
$C_{3}$
N based on the DFT+
$G_{0}$

$W_{0}$
 methodology to obtain an improved value for the band gap and the Bethe-Salpeter approach to calculate the excitonic properties. Several findings of Refs. [21,22] can be interpreted with the help of the results presented in Section 3.

According to the numerical calculations of Ref. [22], the lowest energy direct excitonic state is doubly degenerate and dark. The corresponding electron-hole transitions are located in the vicinity of the 
Γ
 point, and the electron part of the exciton wavefunction is localized on the benzene rings of 
$C_{3}$
N if the hole is fixed on an N atom.

Firstly, we note that there is no 
k·p
 matrix element between the VB and the doubly degenerate CB at the 
Γ
 point (see Appendix A), which indicates that direct optical transitions are forbidden by symmetry. Furthermore, as shown in Figure 3, at the 
Γ
 point, the 
$p_{z}$
 atomic orbitals of the C atoms have a large weight in the CB, CB+1 bands, and the same applies to the 
$p_{z}$
 atomic orbitals of the N atoms in the VB. According to Table A2, the degenerate CB, CB+1 bands correspond to the two-dimensional 
$E_{2u}$
 irreducible representation of 
$D_{6h}$
. One can check that the following linear combinations of the 
$p_{z}$
 atomic orbitals of the C atoms transform as the partners of the 
$E_{2u}$
 irreducible representation:
(7)
$$\begin{array}{cc} \; & \phi_{1}=\frac{1}{\sqrt{6}}\left[p_{z}^{\left(1,C\right)}+\omega p_{z}^{\left(2,C\right)}+\omega^{2}p_{z}^{\left(3,C\right)}\right.\\ \; & \;+\left.p_{z}^{\left(4,C\right)}+\omega p_{z}^{\left(5,C\right)}+\omega^{2}p_{z}^{\left(6,C\right)}\right], \end{array}$$


(8)
$$\begin{array}{cc} \; & \phi_{2}=\frac{1}{\sqrt{6}}\left[p_{z}^{\left(1,C\right)}+\omega^{2}p_{z}^{\left(2,C\right)}+\omega p_{z}^{\left(3,C\right)}\right.\\ \; & \;+\left.p_{z}^{\left(4,C\right)}+\omega^{2}p_{z}^{\left(5,C\right)}+\omega p_{z}^{\left(6,C\right)}\right], \end{array}$$

where 
$\omega =e^{2\pi i/3}$
, and 
$p_{z}^{\left(i,C\right)}$
, 
i=1,2,…,6
 denote the 
$p_{z}$
 orbitals of the six-carbon atoms in the unit cell; see Figure 1. Bloch wavefunctions based on 
$\phi_{1}$
 and 
$\phi_{2}$
 would indeed have a large weight on the benzene rings in each unit cell, and this helps to explain the corresponding finding of Ref. [22].

On the other hand, the 
k·p
 matrix elements are non-zero between pairs of the degenerate VB-1, VB-2 and CB, CB+1 bands; see Table A3. This means that optical transitions are allowed by symmetry, and if circularly polarized light is used for excitation, then 
$\sigma^{+}$
 and 
$\sigma^{-}$
 would excite transitions between different pairs of bands. Some of the higher-energy bright excitonic states found in Ref. [22] should correspond to this transition.

At the *M* point, on the other hand, there is finite matrix element between the VB and the CB—see Equation (Equation 6)—which suggests that optical transitions are allowed by symmetry along the 
Γ−M
 line. Moreover, one can expect that the optical density of states is large going from *M* toward 
Γ
 because the VB and the CB are approximately parallel. Note that the time-reversed states can be found at 
$-q_{x}$
, i.e., on the other side of the 
Γ
 point. Therefore, one can expect that in a zero magnetic field, two degenerate bright excitonic states can be excited, and in the 
k
 space, they are localized on opposite sides of the 
Γ
 point along the 
Γ−M
 lines. This corresponds to the findings of Ref. [22]. In an external magnetic field, which breaks time reversal symmetry, the degeneracy of the two excitonic states would be broken. This is reminiscent of the valley degeneracy breaking for magnetoexcitons in monolayer TMDCs [43,44,45,46,47].

The transition along the 
Γ−M
 line can be excited by a linearly polarized light. For a general direction of the linear polarization with respect to the crystal lattice, transitions along all three 
Γ−M
 directions in the BZ would be excited. However, when the polarization of the light is perpendicular to one of the 
Γ−M
 line, then the interband transitions are excited only in the remaining two 
Γ−M
 “valleys”. In contrast, when circularly polarized light is used for excitation, all three 
Γ−M
 “valleys” are excited.

## 5. Landau Levels

We now consider the Landau-level (LL) spectrum of monolayer 
$C_{3}$
N. Using the 
k·p
 Hamiltonians, one can employ the Kohn–Luttinger prescription; see, for example, Ref. [39]. This means that one can replace the wavenumber 
$q=(q_{x},q_{y})$
 by the operator 
$\hat{q}=\frac{1}{i}\nabla +\frac{e}{\hbar }A$
, where 
e>0
 is the magnitude of electron charge, and 
A
 is the vector potential describing the magnetic field. We expect that this approach should be accurate for low-energy LLs, which are our main interest here. For high magnetic fields and/or high LL indices, *n* more advanced methods may need to be used [48].

In the following, we will use the Landau gauge and 
$A={\left(0,B_{z}x,0\right)}^{T}$
. Since the components of 
$\hat{q}$
 do not commute, the Kohn–Luttinger prescription should be performed in the original nine-band (
Γ
 point) or thirteen-band (*M* point) 
k·p
 Hamiltonians (see Appendices Appendix A and Appendix B) and not in the low-energy effective ones given in Equations (Equation 1)–(Equation 3) and (Equation 4)–(Equation 6), respectively. After the Kohn–Luttinger prescription in the higher-dimensional 
k·p
 model, one can again use the Löwdin partitioning to obtain low-energy effective Hamiltonians by taking care of the order of the non-commuting operators appearing in the downfolding procedure. This approach was used, for example, in the case of monolayer TMDCs to study the valley-degeneracy breaking effect of the magnetic field [49].

### 5.1. Effective Model at the 
Γ
 Point

The low-energy model can be expressed in terms of 
${\hat{q}}_{\pm }={\hat{q}}_{x}\pm i{\hat{q}}_{y}$
. Note, that 
${\hat{q}}_{+}$
 and 
${\hat{q}}_{-}$
 do not commute, so that 
$\left[{\hat{q}}_{-},{\hat{q}}_{+}\right]=\frac{2eB_{z}}{\hbar }$
, where 
$B_{z}$
 is a perpendicular magnetic field:
(9)
$$\begin{array}{c} H_{\mathrm{eff}}^{\Gamma }=H_{0}^{\Gamma }+H^{\Gamma }\left(B_{z}\right)+H_{Z}, \end{array}$$

where 
$H_{0}^{\Gamma }$
 was defined in Equations (Equation 1)–(Equation 3), 
$H_{Z}=\frac{1}{2}g_{e}\mu_{b}B_{z}S_{z}$
 is the Zeeman term, and

(10)
$$\begin{array}{c} H^{\Gamma }\left(B_{z}\right)=\left(\begin{array}{ccc} {\hat{h}}_{11} & 0 & 0\\ 0 & {\hat{h}}_{22} & {\hat{h}}_{23}\\ 0 & {\hat{h}}_{32} & {\hat{h}}_{33} \end{array}\right). \end{array}$$


The operators 
${\hat{h}}_{ij}$
 in 
$H^{\Gamma }\left(B_{z}\right)$
 are defined as follows:
(11)
$${\hat{h}}_{11}=\frac{1}{2m_{vb}^{\Gamma }}\left({\hat{q}}_{+}{\hat{q}}_{-}+e\hbar B_{z}\right).$$


(12)
$$\begin{array}{c} {\hat{h}}_{22}=\left(\alpha_{2}+\alpha_{3}\right){\hat{q}}_{+}{\hat{q}}_{-}+\alpha_{2}\frac{2eB_{z}}{\hbar }-\frac{\hbar eB_{z}}{2m_{e}} \end{array}$$


(13)
$$\begin{array}{c} {\hat{h}}_{23}=-\alpha_{3}{\left({\hat{q}}_{+}\right)}^{2},\;{\hat{h}}_{32}=-\alpha_{3}{\left({\hat{q}}_{-}\right)}^{2}, \end{array}$$


(14)
$$\begin{array}{c} {\hat{h}}_{33}=\left(\alpha_{2}+\alpha_{3}\right){\hat{q}}_{+}{\hat{q}}_{-}+\alpha_{3}\frac{2eB_{z}}{\hbar }+\frac{\hbar eB_{z}}{2m_{e}}. \end{array}$$


Here, 
${\hat{h}}_{11}$
 corresponds to the VB, while the degenerate CB, CB+1 are described by the 
2×2
 block in Equation (Equation 10). One can introduce the creation and annihilation operators 
$a^{†}$
, *a* by 
${\hat{q}}_{-}=\frac{\sqrt{2}}{l_{B}}a$
, 
${\hat{q}}_{+}=\frac{\sqrt{2}}{l_{B}}a^{†}$
, where 
$l_{B}=\sqrt{\hbar /eB_{z}}$
 is the magnetic length. The LLs that are obtained from 
${\hat{h}}_{11}$
 correspond to the usual harmonic oscillator spectrum 
$E_{vb}^{\Gamma }=\hbar \omega_{vb}^{\Gamma }\left(n+1/2\right)$
, where 
$\omega_{vb}=eB_{z}/m_{vb}^{\Gamma }$
 and 
n=0,1,2…
 is a positive integer.

Regarding the LLs of the degenerate CB, CB+1 bands, one can anticipate from Equations (Equation 12)–(Equation 14) that the LL spectrum of this minimal model exhibits an interplay of features known from bilayer graphene and conventional semiconductors. Two eigenstates read

(15)
$$\Psi_{0}=\left(\begin{array}{c} |0\rangle \\ 0 \end{array}\right),\;\Psi_{1}=\left(\begin{array}{c} |1\rangle \\ 0 \end{array}\right),$$

where 
|0〉
 and 
|1〉
 are harmonic oscillator eigenfunctions. The corresponding eigenvalues are 
$E_{0}^{\Gamma }=\alpha_{2}\frac{2eB_{z}}{\hbar }-\frac{\hbar eB_{z}}{2m_{e}}$
 and 
$E_{1}^{\Gamma }=\left(2\alpha_{2}+\alpha_{3}\right)\frac{2eB_{z}}{\hbar }-\frac{\hbar eB_{z}}{2m_{e}}$
, respectively. The eigenstates in Equation (Equation 15) have the same form as the two lowest energy eigenstates of bilayer graphene. However, 
$E_{0}$
 and 
$E_{1}$
 are not degenerate, and they do depend on a magnetic field, unlike in the case of bilayer graphene.

The rest of the eigenvalues can be obtained by using the Ansatz:
(16)
$$\Psi_{n}=\left(\begin{array}{c} a_{1}|n+2\rangle \\ a_{2}|n\rangle \end{array}\right),$$

where 
n=0,1,2⋯
 and 
$a_{1}$
,
$a_{2}$
 are constants. This Ansatz leads to a 
2×2
 eigenvalue equation yielding the energy of two LLs for each *n*. The eigenvalues can be analytically calculated, but the resulting expression is quite lengthy and not particularly insightful.

We plot the first four LLs as a function of the magnetic field in Figure 4. They correspond to 
$E_{0}^{\Gamma }$
 and 
$E_{1}^{\Gamma }$
 given below in Equation (Equation 15) and the two LLs that can be obtained using the Ansatz in Equation (Equation 16) for 
n=0
. For comparison, we also plot the energies of “conventional” LLs 
$E_{n}^{cb}=\hbar \omega_{cb}^{\Gamma }\left(n+\frac{1}{2}\right)$
 and 
$E_{n}^{cb+1}=\hbar \omega_{cb+1}^{\Gamma }\left(n+\frac{1}{2}\right)$
, where 
$\omega_{cb}^{\Gamma }=eB_{z}/m_{cb}^{\Gamma }$
 (
$\omega_{cb+1}^{\Gamma }=eB_{z}/m_{cb+1}^{\Gamma }$
) and the effective mass 
$m_{cb}^{\Gamma }$
 (
$m_{cb+1}^{\Gamma }$
) was defined below Equation (Equation 1)–(Equation 3). One can see that the LLs calculated using Equations (Equation 15) and (Equation 16) are different from the conventional LLs, which indicates the important effect of the interband coupling. In Figure 5, we show the LL energies as a function of the LL index *n* for a fixed magnetic field 
$B_{z}=10$
 T. One can see that for large *n*, the LLs energies obtained from Equation (Equation 16) (magenta dots) run parallel to the conventional LLs (black dots). This means that in this limit, both set of LLs can be described by cyclotron energies 
$\hbar \omega_{cb}^{\Gamma }$
 and 
$\hbar \omega_{cb+1}^{\Gamma }$
, but there is an energy difference between them. However, in the deep quantum regime (
n=0,1
) the two sets of LLs cannot be characterized by the same cyclotron frequencies.

We note, however, that electron–electron interactions may modify the above single particle LL picture, similarly to what has recently been found in monolayer 
${\mathrm{MoS}}_{2}$
 [50]. Namely, the obtained effective masses of the degenerate CB and CB+1 bands are quite large, which means that the kinetic energy of the electrons is suppressed. The importance of the electron–electron interactions can be characterized by the dimensionless Wigner–Seitz radius 
$r_{s}=1/\left(\sqrt{\pi n_{e}}a_{B}^{*}\right)$
. Here, 
$n_{e}$
 is the electron density, 
$a_{B}^{*}=a_{B}\left(\kappa m_{e}/m^{*}\right)$
 is the effective Bohr radius, 
$m^{*}$
 is the effective mass, 
κ
 is the dielectric constant and 
$a_{B}$
 is the Bohr radius. Taking [31] 
κ=5
 and an electron density of 
$n=4\times 10^{12}$


${\mathrm{cm}}^{2}$
, one finds 
$r_{s}=2.35$
 for the heavier band, where 
$m^{*}/m_{e}=0.22$
. This value of 
$r_{s}$
 indicates that electron–electron interactions can be important. Therefore, this material may offer an interesting system to study the interplay of electron–electron interactions and interband coupling.

### 5.2. M Point

We obtain the following low-energy effective Hamiltonian in terms of the operators 
${\hat{q}}_{x}$
 and 
${\hat{q}}_{y}$
:
(17)
$$\begin{array}{c} H_{\mathrm{eff}}^{M}=H_{0}^{M}+H^{M}\left(B_{z}\right)+H_{Z} \end{array}$$

where 
$H_{0}^{M}$
 was defined in Equation (Equation 5) and 
$H^{M}\left(B_{z}\right)$
 corresponds to Equation (Equation 6):
(18)
$$\begin{array}{c} H^{M}\left(B_{z}\right)=\left(\begin{array}{cc} {\hat{h}}_{11} & {\hat{h}}_{12}\\ {\hat{h}}_{21} & {\hat{h}}_{22} \end{array}\right), \end{array}$$

where

(19)
$$\begin{array}{c} {\hat{h}}_{11}=\frac{\hbar^{2}}{2m_{e}}{\hat{q}}_{x}^{2}+\beta_{1}{\hat{q}}_{y}^{2}, \end{array}$$


(20)
$$\begin{array}{c} {\hat{h}}_{12}=\gamma_{21}{\hat{q}}_{x},\;{\hat{h}}_{21}=\gamma_{21}^{*}{\hat{q}}_{x}, \end{array}$$


(21)
$$\begin{array}{c} {\hat{h}}_{22}=\beta_{2}{\hat{q}}_{x}^{2}+\beta_{3}{\hat{q}}_{y}^{2}. \end{array}$$


The off-diagonal elements of 
$H^{M}\left(B_{z}\right)$
 are significantly smaller than the diagonal ones for magnetic fields 
$B_{z}≲15$
 T; therefore, one can again use the Löwdin partitioning to transform them out. Since the band edge at the *M* point is in the VB, in the following, we consider the LLs in this band. By rewriting 
${\hat{q}}_{x}$
 and 
${\hat{q}}_{y}$
 in terms of annihilation and creation operators 
${\hat{q}}_{x}=\frac{1}{\sqrt{2}l_{B}}\left({\hat{a}}^{†}+\hat{a}\right)$
, and 
${\hat{q}}_{y}=\frac{-i}{\sqrt{2}l_{B}}\left({\hat{a}}^{†}-\hat{a}\right)$
, one finds that the LLs for VB are given by

(22)
$$\begin{array}{c} E_{n,vb}=\varepsilon_{vb}+\hbar \omega_{vb}^{M}\left(n+\frac{1}{2}\right)+\frac{1}{2}g_{e}\mu_{b}B_{z}S_{z}. \end{array}$$


Here 
$\omega_{vb}^{M}=\frac{eB_{z}}{2m_{vb}^{*}}$
 is the cyclotron frequency, where 
$m_{vb}=\sqrt{m_{vb,x}m_{vb,y}}$
 and 
$m_{vb,x}$
 and 
$m_{vb,y}$
 refer to the effective masses along the 
Γ
-*M* and *M*-*K* directions; see Table 1.

## 6. Summary

In summary, we derived effective low-energy Hamiltonians for monolayer 
$C_{3}$
N at the 
Γ
 and *M* points of the Brillouin zone. We showed that at the 
Γ
 point, there are no linear-in-
q
 matrix elements between the VB and the degenerate CB bands, which means that optical transition is not allowed. We showed that optical transitions are allowed between the degenerate VB-1, VB-2 and the CB, CB+1 bands at the 
Γ
 point if circularly polarized light is used. We also found that there is a saddle point in the energy dispersion of *M* point. In addition, we suggested that the transition along the 
Γ−M
 line can be excited by linearly polarized light at the *M* point. Moreover, we obtained the Landau-level spectra by employing the Kohn–Luttinger prescription for the 
k·p
 Hamiltonians at the 
Γ
 and *M* points. We pointed out the important effect of the interband coupling on the Landau-level spectrum at the 
Γ
 point. An interesting further direction could be the study of electron–electron interaction effects on the Landau-level splittings, which can be important due to the heavy effective mass.

## Figures and Tables

**Figure 1 nanomaterials-12-04375-f001:**
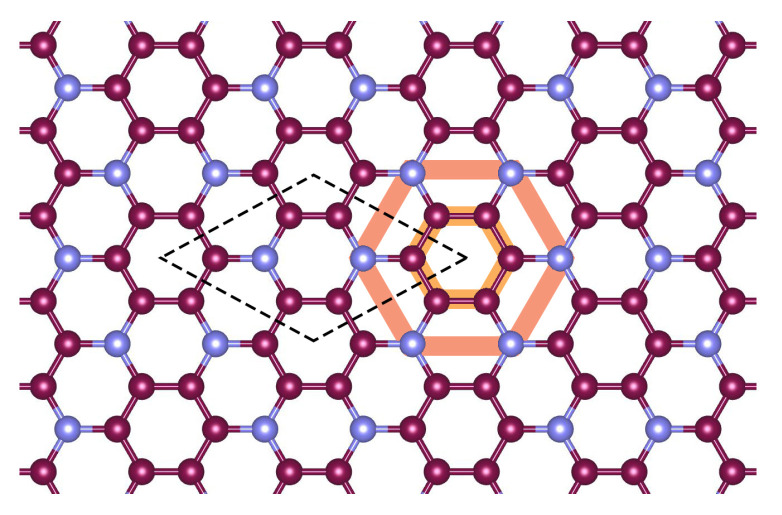
Crystal structure of 
$C_{3}$
N monolayer. Purple and blue circles refer to carbon and nitrogen atoms, respectively. Dashed black lines show the unit cell. The orange line show the hexagonal unit cell, which can be useful for understanding certain optical properties; see Section 4.

**Figure 2 nanomaterials-12-04375-f002:**
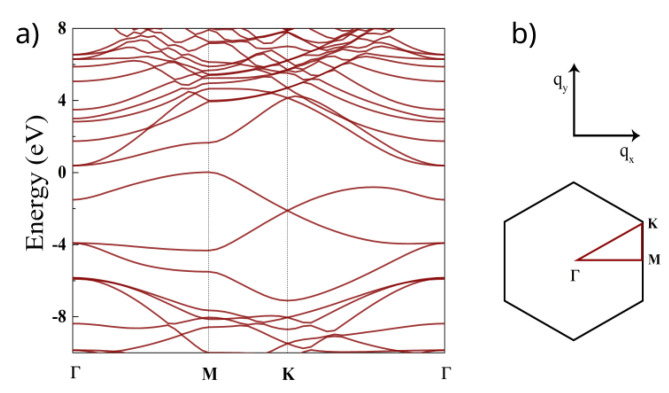
(**a**) DFT band structure calculations for 
$C_{3}$
N along the 
Γ−K−M−Γ
 line in the BZ. (**b**) orientation of the BZ and the high-symmetry points 
Γ
, *K*, *M*.

**Figure 3 nanomaterials-12-04375-f003:**
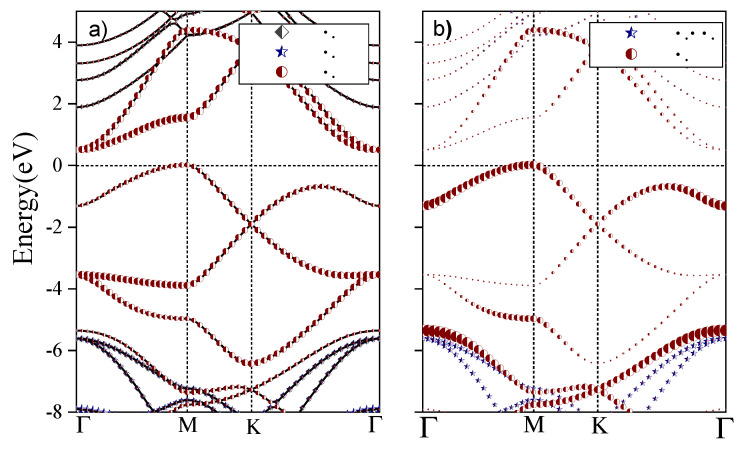
Atomic orbital weight of (**a**) carbon and (**b**) nitrogen atoms in the energy bands of 
$C_{3}$
N monolayer.

**Figure 4 nanomaterials-12-04375-f004:**
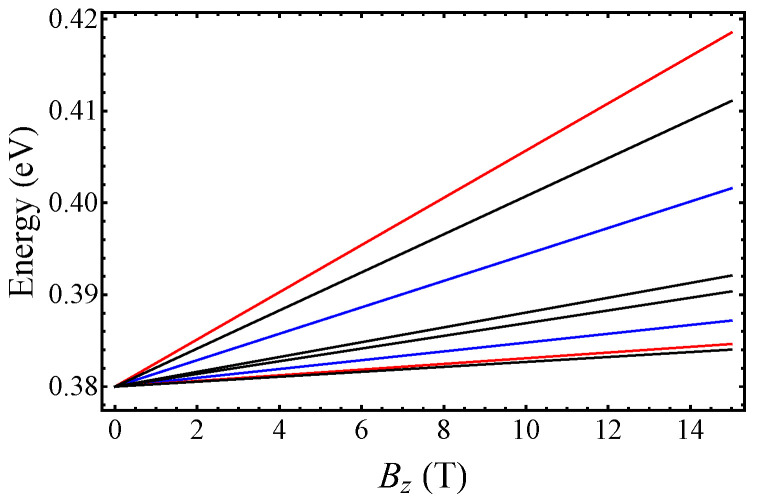
Landau levels in the CB at the 
Γ
 point of the BZ as a function of the out-of-plane magnetic field 
$B_{z}$
. Blue lines show 
$E_{0}^{\Gamma }$
 and 
$E_{1}^{\Gamma }$
, given below in Equation (Equation 15), and red lines indicate the first two LLs that can be obtained from the Ansatz in Equation (Equation 16). Black lines show the “conventional” LLs. See the manuscript.

**Figure 5 nanomaterials-12-04375-f005:**
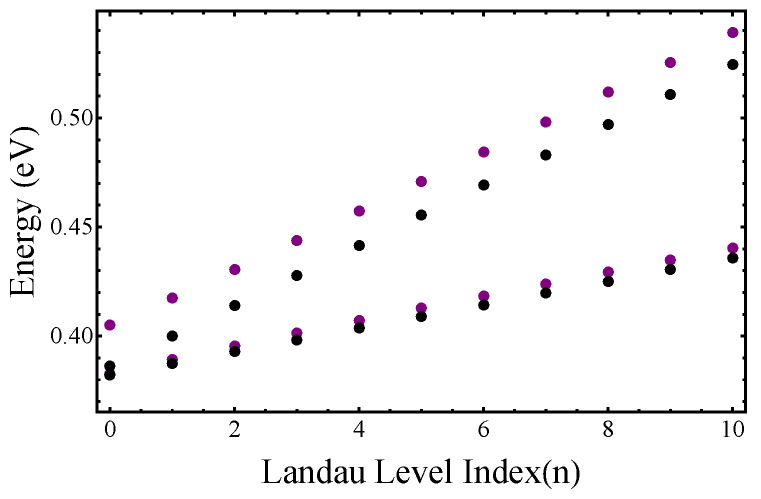
Landau levels in the CB at the 
Γ
 point of the BZ as a function of the LL index n for 
$B_{z}=10$
 T. Magenta dots indicate LLs that can be obtained from the Ansatz in Equation (Equation 16) of the manuscript, and black dots show the “conventional” LLs. See the manuscript.

**Table 1 nanomaterials-12-04375-t001:** Effective masses at the 
Γ
 and *M* points.

	All Directions	*M*– Γ Line	*M*–K Line
$m_{vb}^{\Gamma }/m_{e}$	0.07	-	-
$m_{cb}^{\Gamma }/m_{e}$	0.22	-	-
$m_{cb+1}^{\Gamma }/m_{e}$	0.08	-	-
$m_{vb}^{M}/m_{e}$	-	−0.25	−0.03
$m_{cb}^{M}/m_{e}$	-	−0.19	0.03

## Data Availability

Not applicable.

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
