# Peer review of "Effective Low-Energy Hamiltonians and Unconventional Landau-Level Spectrum of Monolayer C_3_N"

_nanomaterials, 2022, doi:10.3390/nano12244375_

Round 1
Reviewer 1 Report
The article is correct in terms of content. The study covers current scientific issues. The symmetry analysis of the electronic bands was correctly performed. Well calculated Landau 4 level spectrum. Well-chosen literature items.
These are the positive aspects of the study.
Potentials for improvement.
-) Please indicate in the abstract what is the purpose of the article.
-) Please give in the abstract the utilitarian benefits of the study.
-) In the text before the figures, please mention the figure number (e.g. figure 1).
-) I propose not to end the subsection with a table only with the summary text (e.g. subsection 3.1)
-) please end the summary (chapter 6) with text. Directions for future research may be given.
I don't feel qualified to judge about the English language and style.
I recommend publishing the article.
Author Response
We are happy to see that the First Reviewer recommends the publication of the manuscript. We thank the Reviewer for his/her work and comments on the manuscript, which has helped us to improve the manuscript.
She/he has made several suggestions, which we address point-by-point:
- Please indicate in the abstract what is the purpose of the article.
We are not sure what does the Referee mean by this suggestion. For example, the first sentence in the abstract reads “We derive a low-energy effective k p Hamiltonians for monolayer C3N at the Γ and M points of the Brillouin zone…” We think that this indicates quite clearly what is one of our aims in this work.
- Please give in the abstract the utilitarian benefits of the study.
We are not sure what does the “utilitarian benefit” mean. However, we say in the abstract that, for example “Our analysis of the electronic band symmetries helps to better understand several results of recent ab-initio calculations [1,2] for the optical properties of this material”.
- In the text before the figures, please mention the figure number (e.g. figure 1).
We have changed the placement of Figure 2, and as a results, the first mention of Figure 1 in the manuscript text is now closer to the place where Figure 1 appears.
- I propose not to end the subsection with a table only with the summary text (e.g. subsection 3.1)
We thank the Referee for this suggestion, we have changed the placement of Table 1, so that subsection 3.1 ends with text.
- please end the summary (chapter 6) with text. Directions for future research may be given.
We are not exactly sure what does the Referee mean by this suggestion. Section 6 “Summary” contains only text. In the last sentence of Section 6 we also indicate possible future work: “ An interesting further direction could be the study of electron-electron interaction effects on the Landau level splittings...”
Reviewer 2 Report
The paper presents the effective low energy Hamiltonians and unconventional Landau level spectrum of monolayer C3N. According to the reviewer’s opinion, the paper is well-structured and clear. The topic is interesting and falls within the aim of the journal. In addition, the results are well-presented and could be helpful to further develop the same topic. Therefore, the paper can be accepted for publication in the current form.
Author Response
We are happy to see that the Second Reviewer recommends the publication of the manuscript in its present form. She/he finds that the topic is interesting and the results are well-presented. We thank the Reviewer for her/his work and very positive evaluation of our results.
Reviewer 3 Report
The submitted research is well organized. However, one issue, and the most critical, it the concept which is not properly highlighteted. Since it is a purely theoretical article, it is not stressed which are the prons and cons of the method. For example, have the compared their approach with those reported in https://journals.aps.org/prb/abstract/10.1103/PhysRevB.106.155414 ?
Author Response
We thank the Reviewer for her/his work. The Reviewer finds that the research presented in our manuscript is well organized. He raises two questions, which we address below.
1) “... which are the prons and cons of the method?”
The k.p methodology is a well established approach and has been used by several generations of theoretical physicists. We believe that its strengths and weaknesses are known to the community. Discussion of “ prons and cons” of a method is more interesting in the case of less known and less frequently employed approaches, such as the Haydock-Heine-Kelly method discussed in https://journals.aps.org/prb/abstract/10.1103/PhysRevB.106.155414
We note that one of the strengths of the k.p methodology can be seen in Section 5 of our manuscript: it allows to obtain analytical results for the LL energies, which is usually not the case for e.g. tight-binding models.
2) “Have the authors compared their approach with those reported in
https://journals.aps.org/prb/abstract/10.1103/PhysRevB.106.155414 ?”
In the publication cited by the Third Reviewer the authors discuss the use of the Haydock-Heine-Kelly (HHK) method to calculate the Landau level (LL) spectrum in monolayer graphene.
They find, among others, that for low energy LLs the standard approach, which is based on the k.p model of graphene and the minimal coupling substitution to take into account the magnetic field, gives a good agreement with the results obtained from the HHK method. Deviations appear for higher LL index N and stronger magnetic fields.
We expect that the situation should be similar in the case of C3N as well. We note that in our work we considered relatively low LLs (N≤10) and magnetic fields (B≤15T) where we expect that the k.p method based approach should be accurate.
In the manuscript we now explicitly point out that our results are expected to be valid for low energy LLs. We have added the following sentences to the first paragraph of Section 5:
“We expect that this approach should be accurate for low-energy LLs which are our main interest here. For high magnetic fields and/or high LL indices n more advanced methods may need to be used, see https://journals.aps.org/prb/abstract/10.1103/PhysRevB.106.155414 “.
Round 2
Reviewer 3 Report
The revised version looks consistent compared with its first version and can be published as it is